# Interaction of Receptor-Binding Domain of the SARS-CoV-2 Omicron Variant with hACE2 and Actin

**DOI:** 10.3390/cells13161318

**Published:** 2024-08-07

**Authors:** Ai Fujimoto, Haruki Kawai, Rintaro Kawamura, Akira Kitamura

**Affiliations:** 1Laboratory of Cellular and Molecular Sciences, Graduate School of Life Science, Hokkaido University, N21W11, Kita-ku, Sapporo 001-0021, Hokkaido, Japan; fujimoto.ai.n2@elms.hokudai.ac.jp (A.F.);; 2Laboratory of Cellular and Molecular Sciences, Faculty of Advanced Life Science, Hokkaido University, N21W11, Kita-ku, Sapporo 001-0021, Hokkaido, Japan; 3PRIME, Japan Agency for Medical Research and Development, Chiyoda-ku, Tokyo 100-0004, Japan

**Keywords:** COVID-19, spike protein, actin, fluorescence correlation spectroscopy, protein–protein interaction

## Abstract

The omicron variant of severe acute respiratory syndrome coronavirus 2 (SARS-CoV-2) was identified in 2021 as a variant with heavy amino acid mutations in the spike protein, which is targeted by most vaccines, compared to previous variants. Amino acid substitutions in the spike proteins may alter their affinity for host viral receptors and the host interactome. Here, we found that the receptor-binding domain (RBD) of the omicron variant of SARS-CoV-2 exhibited an increased affinity for human angiotensin-converting enzyme 2, a viral cell receptor, compared to the prototype RBD. Moreover, we identified β- and γ-actin as omicron-specific binding partners of RBD. Protein complex predictions revealed that many omicron-specific amino acid substitutions affected the affinity between RBD of the omicron variant and actin. Our findings indicate that proteins localized to different cellular compartments exhibit strong binding to the omicron RBD.

## 1. Introduction

Coronavirus disease 2019 (COVID-19) is caused by the positive-strand RNA virus, severe acute respiratory syndrome coronavirus 2 (SARS-CoV-2). The global COVID-19 pandemic caused by the prototype SARS-CoV-2 virus (Wuhan-Hu-1) began toward the end of 2019 [1]. COVID-19 rapidly spread worldwide, leading to the emergence of different variants due to the large number of infected people. Omicron (B.1.1.529), a SARS-CoV-2 variant, was first reported in South Africa in November 2021 [2]. Following the original variant B.1.1.529, several subvariants have emerged (BA.1–5) [2,3]. Spike glycoproteins (S proteins) on the virus surface recognize human cell receptors, such as the human angiotensin-converting enzyme 2 (hACE2), and mediate membrane fusion between the virus and human cells [4,5]. The S protein is divided into S1 and S2 subunits during viral infections, with the S1 subunit containing the receptor-binding domain (RBD) [6,7]. Dissociation constant (*K*_d_) is a specific type of equilibrium constant that measures the propensity of a molecular complex to dissociate reversibly into smaller components, and it is often used to quantify and express the interaction strength. As the physiological functions of all life including cells and viruses are based on non-covalent biomolecular interactions [8], the biomolecular interactions between virus-derived biomolecules and intracellular or cell surface biomolecules can provide important evidence for viral infection and replication. The S protein and RBD of SARS-CoV-2 interact with hACE2, with *K*_d_ of several nanomolars [9,10]. Variant-specific missense mutations are generally few in the viral proteins but relatively abundant in the S protein [11]. The omicron variant harbors heavy amino acid substitutions in the spike protein compared to previous variants [12]. As the affinity between cell receptors and S protein of the virus promotes infection and transmission, the variant-specific affinity of hACE2 warrants further investigation. Various biochemical and in silico studies have reported the mechanisms by which amino acid substitutions affect the affinity of the omicron variants for hACE2 [9,10,13,14,15]. T478K, Q493K, and Q498R significantly contribute to the increased affinity between RBD and hACE2. The E484A substitution elicits escaping effects from the neutralization escape of beta, gamma, and mu variants. T478K, Q493K, Q498R, and E484A substitutions contribute to weaken the affinity between RBD and neutralization monoclonal antibodies [12]. However, the findings are inconclusive. Fluorescence correlation spectroscopy (FCS) and two-color fluorescence cross-correlation spectroscopy (FCCS) are widely used to study the molecular interactions with single-molecule sensitivity in solutions and live cells [16,17,18]. Moreover, FCS can be used to analyze the intermolecular interactions, even in cell lysates and under low-purification conditions. In this study, we aimed to compare the interactions between fluorescent protein-tagged recombinant hACE2 and RBDs of both the prototype and omicron variants (Wh1 and Omic, respectively) of SARS-CoV-2 using FCCS.

Many studies have investigated the affinity of S protein for cellular surface proteins, the folding process in the endoplasmic reticulum (ER), and secretory pathways involved in the production and assembly of new viral components. However, the interactions of S protein with intracellular proteins in cellular subcompartments, which are distant from the protein synthesis pathways, are more diverse than initially expected [19,20]. Components of focal adhesion, filopodium, ER membrane, and mitochondrial membrane are the potential interactors of the S protein [19]. If the S protein is expressed in the secretory pathway and cytoplasm, it possibly causes various interactome changes, affecting the immune responses, cell viability, and viral infectivity. In fact, the ORF2 protein of hepatitis E virus, which undergoes glycosylation in the ER lumen, is retrotranslocated to the cytoplasm, even when it is not subject to ER-associated protein degradation [21]. Various virus-derived proteins have been reported to bind to cellular cytoplasmic components such as the actin cytoskeleton [22,23], and the significance of the SARS-CoV-2-actin interaction has been discussed [24]. These findings suggest that the interactions between cytoplasmic protein components and viral proteins may potentially influence viral infection, replication, and cytotoxicity. Here, we demonstrated the enhanced binding affinity of omicron RBD for hACE2 compared to that of prototype RBD. Additionally, we identified cytoplasmic actin as a potential interactor of omicron RBD.

## 2. Materials and Methods

### 2.1. Preparation of Plasmid DNA

Expression plasmids (pER-mCherry-RBD^Wh1^ and phACE2-eGFP) for protein purification were prepared as previously described [25]. As previously reported [25], the synthetic oligonucleotides for RBD^Omic^ carrying mutations in the B.1.1.529 variant were annealed and inserted into pER-mCherry-C1 (pER-mCherry-RBD^Omic^). For fluorescence imaging, mCherry-tagged ER-sorting signal-lacking RBD plasmids were prepared via vector backbone exchange (pmCherry-RBD^Wh1^ and pmCherry-RBD^Omic^). The plasmid for GFP-β-actin expression was modified as a cDNA-encoding fluorescent tag of YFP-β-actin (TaKaRa-Clontech, Shiga, Japan) and substituted with that encoding eGFP (peGFP-actin).

### 2.2. Protein Purification

Recombinant proteins were expressed in murine neuroblastoma Neuro2a cells (CCL-131; ATCC, Manassas, VA, USA) and purified as previously described [25].

### 2.3. Immunofluorescence and Confocal Microscopy

For confocal imaging, pmCherry-RBD^Wh1^ or pmCherry-RBD^Omic^ (0.15 μg) and peGFP-actin (0.05 μg) were transfected into HeLa cells (RCB0007; RIKEN BRC, Ibaraki, Japan) using Lipofectamine 2000 (Thermo Fisher, Waltham, MA, USA) in a cover-glass chamber (5222-004; IWAKI, Shizuoka, Japan), according to the manufacturer’s protocol. After incubation for 24 h, live cells were observed using confocal microscopy. For immunofluorescence, HeLa cells expressing mCherry-RBD^Wh1^ or mCherry-RBD^Omic^ were fixed with 4% paraformaldehyde and stained with phalloidin-iFluor 488 (Cayman, Ann Arbor, MI, USA), as previously described [26]. Cell images were acquired using the confocal laser scanning microscope (LSM510 META; Carl Zeiss, Jena, Germany) with a C-Apochromat 40×/1.2 NA Korr UV-VIS-IR water immersion objective (Carl Zeiss).

### 2.4. Mass Spectrometry (MS) Analysis

For MS analysis, purified proteins were separated via sodium dodecyl sulfate-polyacrylamide gel electrophoresis (SDS-PAGE), followed by silver staining (423413; Cosmo Bio, Tokyo, Japan). The bands of interest were cut and analyzed using matrix-assisted laser desorption/ionization–time-of-flight (MALDI-TOF) MS. Then, peptide mass fingerprinting was performed by Genomine (Pohang, Republic of Korea).

### 2.5. Western Blotting

After SDS-PAGE, the protein-transferred PVDF membranes (GE Healthcare Life Sciences, Chicago, IL, USA) were blocked with 5% skimmed milk in PBS-T. Here, anti-GFP HRP-DirecT (#598-7; MBL, Nagano, Japan), anti-mCherry (#Z2496N; TaKaRa, Shiga, Japan) anti-β-Actin (#bs-0061R; Bioss, Beijing, China), anti-γ-Actin (#11227-1-AP; Proteintech, Rosemont, IL, USA), and horseradish peroxidase-conjugated anti-rabbit IgG (#111-035-144; Jackson ImmunoResearch, West Grove, PA, USA) antibodies were used. All primary antibodies were diluted with the CanGet Signal Immunoreaction Enhancer Solution 1 (TOYOBO, Osaka, Japan), whereas the secondary antibody was diluted with 5% skim milk in PBS-T. The dilution ratio of all solutions containing primary and secondary antibodies was 1:1000. Chemiluminescent signals were measured using the ChemiDoc MP Imager (Bio-Rad, Hercules, CA, USA).

### 2.6. FCCS

FCCS was performed using the LSM510 META + ConfoCor 3 system (Carl Zeiss) with a C-Apochromat 40×/1.2 NA Korr UV-VIS-IR water immersion objective (Carl Zeiss), as previously reported [25]. The relative cross-correlation amplitude (RCA) was calculated as follows:$$RCA:=\frac{G_{c}(0)-1}{G_{r}(0)-1}=\frac{N_{c}}{N_{g}}$$
where *G*_r_(0) is the amplitude of the autocorrelation function of mCherry channel at *τ* = 0, *G*_c_(0) is the amplitude of the cross-correlation function (CCF) between eGFP and mCherry channel at *τ* = 0, and *N*_g_ and *N*_c_ are the mean numbers of eGFP-fluorescent and interacting molecules, respectively. Count per molecule (CPM) was determined by dividing the mean fluorescence intensity by the mean number of fluorescent molecules.

### 2.7. Protein Complex Prediction

Protein complex of the omicron variant and β-actin was predicted using ColabFold v.1.5.5 [27]. The default MSA setting, mmseqs2_uniref_env, was used for prediction, without AMBER force-field relaxation and templates. The number of cycles was set to three. The five predicted structures were ranked using the pLDDT and pTM scores, and the predicted structure with the highest score was adopted. Then, positions of amino acids were illustrated using PyMol 2.5.0 (Schrödinger, Inc., New York, NY, USA).

### 2.8. Statistical Analyses

Statistical analyses were conducted using one-way analysis of variance followed by Tukey’s significant difference post hoc test with the Origin Pro 2024 software (OriginLab Corp., Northampton, MA, USA). *p*-value less than 0.05 is considered to be statistically significant, in which case the null hypothesis should be rejected.

## 3. Results and Discussion

### 3.1. Increased Affinity between RBD of the Omicron Variant and hACE2

Purification of the expressed recombinant ER-mCherry-RBD^Wh1^, ER-mCherry-RBD^Omic^, and hACE2-eGFP expressed in Neuro2a cells was confirmed via Western blotting with fluorescent tags and silver staining of SDS-PAGE gels (Figure 1a,b). A single band of the purified protein was observed via Western blotting (Figure 1a). Although some cellular protein contaminants were observed, similar to a previous report [25], the band patterns were similar for ER-mCherry-RBD^Wh1^, ER-mCherry-RBD^Omic^, and hACE2-eGFP (Figure 1b), suggesting that most contaminants were non-specific protein contaminants. As high purity is not necessary for FCCS measurements, CCF of a mixture of ER-mCherry-RBD and hACE2-eGFP was measured. The amplitude of CCF between mCherry-RBD^Omic^ and hACE2-eGFP was higher than that between mCherry-RBD^Wh1^ and hACE2-eGFP (Figure 1c). The RCA between mCherry-RBD and hACE2-eGFP was significantly high, with the RCA being higher with omicron RBD than with prototype RBD (Figure 1d). Therefore, RBD carrying the omicron variant mutation showed a higher affinity for hACE2 than the prototype. RBD was considered to be a monomer. When the fluorescent brightness of a single molecule (CPM) was evaluated using FCS, the CPMs did not change in all samples compared to those in the eGFP and mCherry monomers as controls (Figure 1e,f), indicating that all samples were monomers, even if they interacted, with no aggregate formation. Therefore, high RCA of the omicron RBD for hACE2 was not due to artificial oligomerization of these proteins. Both omicron and prototype RBDs share highly similar binding properties with hACE2; however, a specific group of RBD residues (S496, R498, and Y501) in omicron variant contribute to significant interactions between RBD and hACE2 [12,13,15]. Various in silico and biochemical studies have reported the affinity of omicron RBD/S protein for ACE2 [9,10,13,14]. Consistent with previous reports, our FCCS results support the high affinity of the omicron RBD for hACE2.

### 3.2. Co-Precipitation of β- and γ-Actin with Omicron RBD

FCCS can measure protein–protein interactions with small volumes (~microliters), thereby eliminating the need for large-scale cell cultures. This may be a reason for the lack of highly purified RBDs. However, this is also a condition for the co-precipitation of proteins strongly binding to RBDs, such as in immunoprecipitation. Silver staining patterns after SDS-PAGE of purified RBD proteins for specific binding proteins of the omicron variant revealed a characteristic ~40 kDa band (Figure 2a). MALDI-TOF-MS analysis revealed that this band contained β- and γ-actin. Western blotting using anti-β- and anti-γ-actin antibodies confirmed that the omicron RBD-specific precipitated band contained β- and γ-actin, which were absent in prototype RBD (Figure 2b,c). As mCherry-labeled RBDs were expressed in the ER, binding of cytoplasmic actin to RBD may occur in the cell lysate. Previous interactome analysis using the S protein ORF sequence of the Wuhan prototype as bait does not identify actin as a major binding partner [20,28,29]. Our findings are consistent with the previous reports.

### 3.3. Cytoplasmic RBD^Omic^ Does Not Localize to Actin Filaments

As actin is a major component of the cytoskeleton, we used confocal fluorescence microscopy to determine whether cytoplasm-expressed mCherry-RBD binds to actin filaments. To observe the actin filaments, we used HeLa cells for microscopy. Actin filaments were visualized using transiently expressed eGFP-β-actin. Although actin filaments were attached to the plasma membrane and stress fibers, no co-localization of mCherry-RBD^Omic^ and mCherry-RBD^Wh1^ to the actin filaments was observed (Figure 3a). Furthermore, fluorescent signals of eGFP-β-actin localized to the non-filament cytoplasmic space and mCherry-RBDs were not colocalized (Figure 3a). Next, to observe the actin filaments, green fluorescent dye-labeled phalloidin was used for mCherry-RBD-expressing cells. However, no co-localization of mCherry-RBD^Omic^ and mCherry-RBD^Wh1^ to the actin filaments was observed (Figure 3b). These results suggest that mCherry-RBD^Omic^ does not bind to stationary actin, but it likely binds to diffuse actin (G-actin) in cells. Although FCCS is a valuable method for detecting molecular interactions in live cells, the analysis of the interaction between mCherry-RBD and dynamic G-actin in live cells has not been achieved due to the significant photobleaching of mCherry and the difficulty, specifically, in the fluorescence labeling of G-actin. Therefore, we have not yet analyzed the interaction between mCherry-RBD and dynamic G-actin in live cells. This issue should be resolved in the future.

### 3.4. Binding of RBD to Actin and Its Physiology

To identify the key amino acids involved in the interaction between RBD^Omic^ and β-actin, we predicted the structure of their protein complex using ColabFold (Figure 4a). Of the total 16 amino acid mutations specific for the omicron variant, 9 amino acids (K417N, G446S, T478K, E484A, Q493R, G496S, Q498R, N501Y, and Y505H) were clustered within 6 Å from the actin surface (Figure 4b). Among the no-missense mutations in omicron RBD (N440K, N501Y, and S477N), the N501Y mutation was located at 5 Å, whereas N440K and S477K were located approximately 15 Å from the actin surface. Among the various missense mutations unique to the omicron variant, those that have not been evaluated for ACE2 binding or viral infectivity may impact the binding affinity. Further investigation is needed due to the presence of amino acids (T478K, Q493K, and Q498R) that enhance RBD-hACE2 affinity, potentially suggesting competitive interaction between actin and hACE2.

S protein, including the RBD, is translocated into the ER lumen immediately after synthesis of the nascent polypeptide chain, preventing its interaction with cytoplasmic actin. In contrast, the ORF2 protein of hepatitis E virus is retrotranslocated to the cytoplasm [21]. Similar to the ORF2 protein of hepatitis E virus, the S protein might also be detected in the host cytoplasm. If synthesized S protein leaks into the cytoplasm, it can interact with actin. The lack of co-localization fluorescence signals observed via confocal microscopy (Figure 3) may be due to rapid changes in actin dynamics in live cells caused by ATP hydrolysis and phosphorylation [22]. These dynamic changes may be less pronounced as ATP concentrations are diluted in cell lysates. In live cells, various regulators of actin dynamics may influence this process. Whether the S protein is expressed in or retrotranslocated to the cytoplasm remains unknown. Furthermore, if it is localized to the cytoplasm, whether the binding of S protein suppresses the actin dynamics and cellular functions or supports virus synthesis also remains unclear.

## 4. Conclusions

In conclusion, we demonstrated that the omicron RBD exhibited higher affinity for hACE2 compared to the prototype RBD using FCCS. Our findings highlight the feasibility of detecting the interactions between the RBD/S protein and human viral receptors, such as hACE2, even with the emergence of different variants. Moreover, our findings suggest FCCS as a rapid validation method for neutralizing antibodies and small-molecule drugs useful to inhibit their interactions. Notably, this study revealed cytoplasmic actin as a strong interactor of omicron RBD. However, the specific roles of S protein in the cytoplasm warrant further investigations.

## Figures and Tables

**Figure 1 cells-13-01318-f001:**
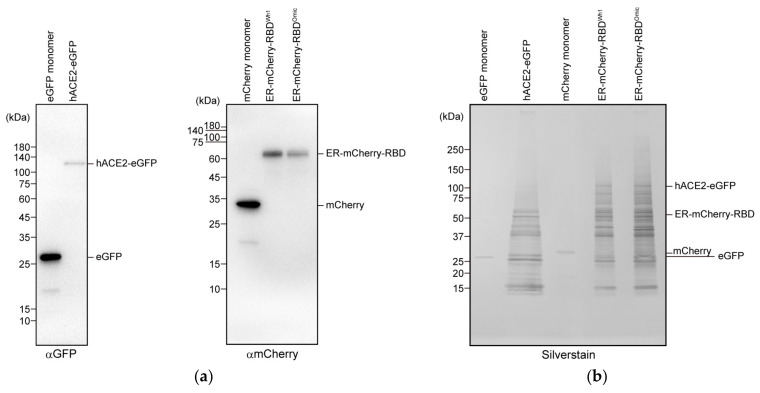
Interaction analysis between angiotensin-converting enzyme 2 (ACE2) and receptor-binding domain (RBD) of omicron/prototype using fluorescence cross-correlation spectroscopy (FCCS). (**a**) Western blotting of the purified recombinant eGFP monomer, hACE2-eGFP, mCherry monomer, ER-mCherry-RBD^Wh1^, and ER-mCherry-RBD^Omic^ using anti-GFP and anti-mCherry antibodies (*left* and *right*, respectively). Uncropped and unedited blots were provided in Appendix A. (**b**) Sodium dodecyl sulfate-polyacrylamide gel electrophoresis (SDS-PAGE) gel followed by silver staining of all purified samples shown in (**a**). Numbers on the left side of the gel image indicate the positions of the molecular weight markers. (**c**) Typical normalized cross-correlation functions ([*G*_c_(τ)−1/*G*_c_(0)−1]) of the mixtures of purified mCherry monomers (Ctrl; gray), ER-mCherry-RBD^Wh1^ (Wh1; green), and ER-mCherry-RBD^Omic^ (Omic; magenta) with hACE2-eGFP. *X*-axis shows the lag time (τ). (**d**) Relative cross-correlation amplitude (RCA) of the indicated two fluorescent color mixtures. (**e**) Counts per molecule (CPM) of eGFP-tagged proteins via FCCS. (**f**) CPM of mCherry-tagged proteins via FCCS. (**d**–**f**) Mo indicates the GFP or mCherry monomer. Bars indicate the mean ± standard error (SE). Dots indicate the independent values. *** *p* < 0.001; NS, not significant (*p* ≥ 0.05). The source data for the graphs (**c**–**f**) are provided in Appendix A.

**Figure 2 cells-13-01318-f002:**
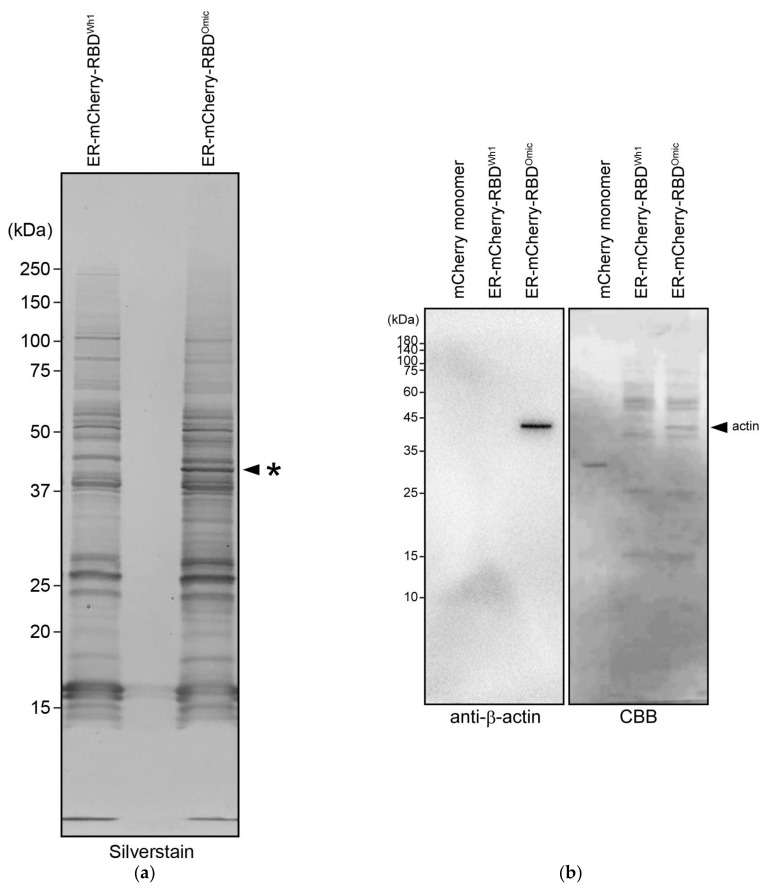
Co-precipitation of beta- and gamma-actin with omicron RBD. (**a**) SDS-PAGE gel followed by silver staining of purified ER-mCherry-RBD^Wh1^ and ER-mCherry-RBD^Omic^. Asterisk indicates the band specifically co-precipitated with the omicron variant. (**b**,**c**) Western blotting of the recombinant mCherry monomer, ER-mCherry-RBD^Wh1^, and ER-mCherry-RBD^Omic^ using anti-β-actin and anti-γ-actin antibodies (**b**,**c**, respectively). Uncropped and unedited blots were provided in Appendix A. Images on the right show the Coomassie brilliant blue (CBB)-stained membranes after antibody detection. Arrowheads indicate the position of β- and γ-actin.

**Figure 3 cells-13-01318-f003:**
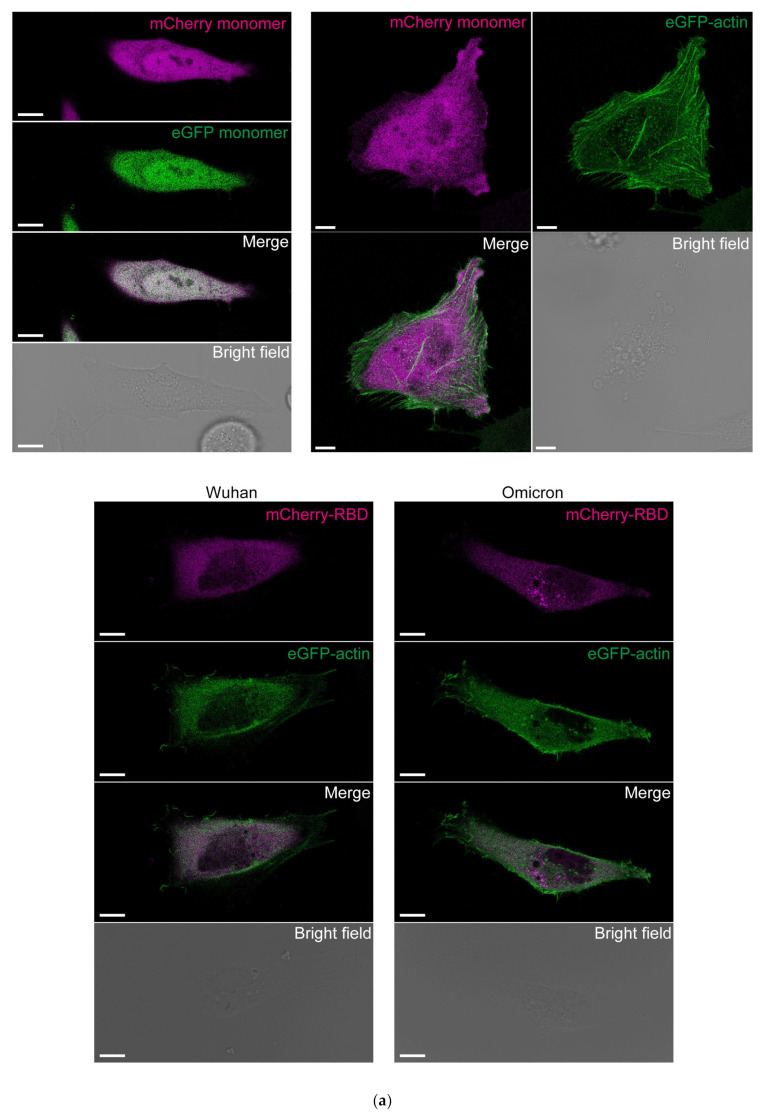
Confocal fluorescence images of actin in HeLa cells expressing mCherry-tagged RBD. (**a**) Fluorescence images of HeLa cells expressing ER-mCherry-RBD^Wh1^ or ER-mCherry-RBD^Omic^ (mCherry-RBD; magenta) with eGFP-β-actin (eGFP-actin; green). Bar = 10 μm. eGFP and mCherry monomers-expressing HeLa cells were used as a control (*upper left*). mCherry monomers- and eGFP-actin-expressing cells were also a control (*upper right*). (**b**) Fluorescence images of HeLa cells expressing mCherry monomers, ER-mCherry-RBD^Wh1^, or ER-mCherry-RBD^Omic^ (mCherry-RBD; magenta) stained with phalloidin-iFluor 488 (Phalloidin; green) and Hoechst 33342 for the nucleus (Hoechst; cyan). Bar = 10 μm.

**Figure 4 cells-13-01318-f004:**
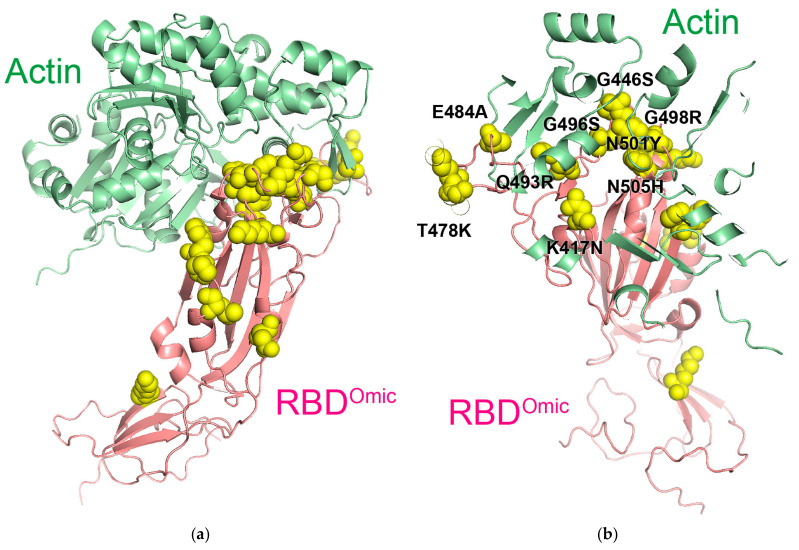
Key amino acids in the predicted complex of omicron RBD and β-actin. (**a**) Predicted protein complex of the omicron RBD (RBD^Omic^; light magenta) and β-actin (light green). Yellow spheres indicate the missense mutations in the omicron RBD. (**b**) Enlarged view of the complex of omicron RBD and actin with higher transparency. Black letters indicate the clustered missense amino acids in the RBD within 6 Å from the actin surface.

## Data Availability

The source data for the graphs were available in Appendix A. Uncropped and unedited blots were provided as Appendix A.

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
