# Peer review of "Interaction of Receptor-Binding Domain of the SARS-CoV-2 Omicron Variant with hACE2 and Actin"

_cells, 2024, doi:10.3390/cells13161318_

Round 1

Reviewer 1 Report

Comments and Suggestions for Authors

The manuscript presents a very interesting topic; however, the narrative is far from being clear and lacks of crucial literature, such as "Kloc, Malgorzata, Ahmed Uosef, Jarek Wosik, Jacek Z. Kubiak, and Rafik M. Ghobrial. 2022. “Virus Interactions with the Actin Cytoskeleton-What We Know and Do Not Know about SARS-CoV-2.” Archives of Virology 167 (3): 737–49.". 

The main flaw of the paper is that from the title, the objective seems "actin", while it relies mainly on the RBD interaction with ACE2 and includes also actin.

The introduction is poor as well, since it is not well explained how it is crucial not to look only at ACE2 interaction, and in explaining the nature of interactions.

Author Response

The manuscript presents a very interesting topic; however, the narrative is far from being clear and lacks of crucial literature, such as "Kloc, Malgorzata, Ahmed Uosef, Jarek Wosik, Jacek Z. Kubiak, and Rafik M. Ghobrial. 2022. “Virus Interactions with the Actin Cytoskeleton-What We Know and Do Not Know about SARS-CoV-2.” Archives of Virology 167 (3): 737–49.".

[Response]

Thank you for your suggestion. As we did not include this important paper, we cite it with substantial introduction content (ref. 24; p. 2, L. 79).

The main flaw of the paper is that from the title, the objective seems "actin", while it relies mainly on the RBD interaction with ACE2 and includes also actin.

[Response]

The reviewer is correct in pointing this out. The title of this manuscript has been changed as follows to clarify that the article is not only about the interaction with actin but also with hACE2.

“Interaction of Receptor-binding Domain of the SARS-CoV-2 Omicron Variant with hACE2 and Actin”

The introduction is poor as well, since it is not well explained how it is crucial not to look only at ACE2 interaction, and in explaining the nature of interactions.

[Response]

Thank you for your careful reading and important suggestion. We modified the introduction.

Reviewer 2 Report

Comments and Suggestions for Authors

Summary

In this manuscript the binding affinity of the SARS-CoV-2 spike protein to hACE2 and actin is presented. The omicron variant showed stronger affinity to hACE2 than the Wuhan original, whereas the latter was found to lack binding to beta- and gamma-actin. Overall, the results are clearly presented and described, however, it would improve the manuscript to add comparisons to findings of other studies.

Revision points

  • Literature search for example comparing with interactome studies the binding of the spike protein with host factors is missing, e.g.:

    • https://www.sciencedirect.com/science/article/pii/S2772892722000657

    • https://www.nature.com/articles/s41587-022-01474-0#data-availability

    • https://www.nature.com/articles/s41586-020-2286-9

  • How do the key amino acids of the spike protein compare with other studies? What functional role does this association involve?

  • lines 226-241: This whole section needs paraphrasing to get clearer. E.g.:

    • lines 227ff: "However,..." I do not understand the consequence. Would you like to express, that the spike protein might be found in the host cytoplasm, as seen for the ORF2 of hepatitis E virus? Please rephrase this sentence for clarification.

  • A language check would also be helpful. More definite articles could be set.

  • Original western blots in Fig. S1 and Fig. S2 look different than to corresponding Fig. 1A. Please check.

Comments on the Quality of English Language

A language check would also be helpful. More definite articles could be set.

Author Response

Summary

In this manuscript the binding affinity of the SARS-CoV-2 spike protein to hACE2 and actin is presented. The omicron variant showed stronger affinity to hACE2 than the Wuhan original, whereas the latter was found to lack binding to beta- and gamma-actin. Overall, the results are clearly presented and described, however, it would improve the manuscript to add comparisons to findings of other studies.

Revision points

Literature search for example comparing with interactome studies the binding of the spike protein with host factors is missing, e.g.:

https://www.sciencedirect.com/science/article/pii/S2772892722000657

https://www.nature.com/articles/s41587-022-01474-0#data-availability

https://www.nature.com/articles/s41586-020-2286-9

[Response]

The three papers presented by this reviewer all use the ORF of the S protein of Wuhan prototyhpe as the bait of the interactome. In out study, the interaction between Wuhan prototype and actin was not detected. Therefore, our results are consistent with these reports. This is added in the main text (p. 6, L. 201–203).

How do the key amino acids of the spike protein compare with other studies? What functional role does this association involve?

[Response]

According to this suggestion, we added a sentence in the discussion section and an explanation in the introduction (p. 2, L. 5458 & p. 10, L. 247250).

lines 226-241: This whole section needs paraphrasing to get clearer. E.g.:

lines 227ff: "However,..." I do not understand the consequence. Would you like to express, that the spike protein might be found in the host cytoplasm, as seen for the ORF2 of hepatitis E virus? Please rephrase this sentence for clarification.

[Response]

Thank you for your careful reading. We rephrased the sentences (p. 10, L. 256265).

A language check would also be helpful. More definite articles could be set.

[Response]

Thank you for your kind comments. This manuscript has edited by a native speaker. The description is written in Acknowledgments (L. 293).

Original western blots in Fig. S1 and Fig. S2 look different than to corresponding Fig. 1A. Please check.

[Response]

Thank you for your careful check of our supplemental figures. They appear different because contrast adjustment is not performed in the raw data (Figures S1/S2).

Reviewer 3 Report

Comments and Suggestions for Authors

The manuscript entitled " Spike Protein of the SARS-CoV-2 Omicron Variant Interacts with Actin" by Ai Fujimoto et al. is an experimental study that employs two-color fluorescence cross-correlation spectroscopy (FCCS) and other approaches to investigate the interaction between the RBD and ACE2 for both Wild-type (WT) and Omicron stains. They found that Omicron RBD interacts with ACE2 more efficiently and tightly than WT RBD. Additionally, they showed that Omicron RBD binds specifically to β- and γ-actin. The study is an interesting and important topic.

The manuscript is well-written and appropriately designed, the methods are clearly described, the results are well-presented, and the discussion is based on the results. Still, it is required more related work to enhance the conclusion further. Therefore, I think this manuscript should be a good addition to the literature in this field. For these reasons, I find this manuscript publishable after minor revisions.

Minor comments:

1)      The sentence in lines 15-16 should be revised.

2)      All supplementary materials must be mentioned in the main text.

3)      Some speculations or evidence, such as those on line 153, require additional references. For example, see and cite the following references:

https://pubs.acs.org/doi/10.1021/acs.jpclett.2c00423

https://doi.org/10.3389/fimmu.2021.830527

4)      Limitations of the work should be mentioned in the result section.

Comments on the Quality of English Language

There are some typos so a minor English edit is required.

Author Response

The manuscript entitled " Spike Protein of the SARS-CoV-2 Omicron Variant Interacts with Actin" by Ai Fujimoto et al. is an experimental study that employs two-color fluorescence cross-correlation spectroscopy (FCCS) and other approaches to investigate the interaction between the RBD and ACE2 for both Wild-type (WT) and Omicron stains. They found that Omicron RBD interacts with ACE2 more efficiently and tightly than WT RBD. Additionally, they showed that Omicron RBD binds specifically to β- and γ-actin. The study is an interesting and important topic.

The manuscript is well-written and appropriately designed, the methods are clearly described, the results are well-presented, and the discussion is based on the results. Still, it is required more related work to enhance the conclusion further. Therefore, I think this manuscript should be a good addition to the literature in this field. For these reasons, I find this manuscript publishable after minor revisions.

[Response]

Thank you for your important suggestions. Based on the comments by the reviewer, we revised the manuscript. Specific responses are described below.

Minor comments:

1)      The sentence in lines 15-16 should be revised.

[Response]

Thank you for your careful reading of our manuscript. Since the introduction of the abstract was missing as the reviewer’s suggestion, we revised it (p. 1, L. 1619).

2)      All supplementary materials must be mentioned in the main text.

[Response]

We mention all supplementary materials in the figure legends (p. 5, L. 179 & 189; p. 7, L. 208209).

3)      Some speculations or evidence, such as those on line 153, require additional references. For example, see and cite the following references:

https://pubs.acs.org/doi/10.1021/acs.jpclett.2c00423

https://doi.org/10.3389/fimmu.2021.830527

[Response]

According to the suggestions, we add the references above on line 153 in the previous edition and the introduction (ref. 12 at L. 50 & 58; ref. 15 at L. 54).

4)      Limitations of the work should be mentioned in the result section.

[Response]

According to this suggestion, we mentioned the limitation of detecting the interaction between mCherry-RBD and actin in live cells using FCCS (p. 7, L. 223229).

Comments on the Quality of English Language

There are some typos so a minor English edit is required.

[Response]

We carefully checked the manuscript and modified the typo.

Round 2

Reviewer 1 Report

Comments and Suggestions for Authors

The final version of the manuscript fully accomplishes for the requirements to be published.